# "*Vergis Mein Nit*"—Connectedness and Commemoration through Rings in the 16th Century

**Romina Ebenhöch**

Museum of the Convent of St. John—UNESCO World Heritage, 7537 Müstair, Switzerland;
romina.ebenhoech@muestair.ch

**Abstract:** By the end of the 16th century, finger rings in reverse glass painting technique became increasingly popular in Europe. Often, they are used in the context of signet rings with the monogram together with the coat of arms of its beholder depicted on the glass bezel. The following paper concentrates on nine finger rings of this group. Instead of an actual coat of arms though, these finger rings carry the device V(G)MN or FGMN (for-get-me-not) accompanied by a depiction of little blue forget-me-not flowers as the coat of arms. By collecting and describing the so far existing material, the paper aims to contextualize the use and function of the finger rings with the symbol of the forget-me-not flower in the fields of love, friendship and faith. Furthermore, it links the symbol of the for-get-me-not on finger rings and the imperative power of the written letters V(G)MN or FGMN to its tradition in German literature and texts.

**Keywords:** 16th century; finger rings; forget-me-not; love and friendship; jewelry; early modern era; reverse glass painting; apotropaic

## 1. Introduction

In the collection of the Vitromusée in Romont in Switzerland is a little finger ring dating to around 1575 (Figure 1). At first glance, the ring with a coat of arms on the bezel seems to be made in the tradition of signet rings typical of the 16th century. Taking a closer look, the ring bezel offers an interesting twist. Instead of initials, one can read the device VGMN (Vergissmeinnicht) accompanying the depiction of little blue forget-me-not flowers as a coat of arms. How are these objects to be understood and contextualized? The following contribution arises from a consideration of how the act of commemoration appears in image and language, particularly in medieval and early modern jewelry. The focus of the study is on the imperative linguistic competence of the forget-me-not and how the filigree flower's figurative linguistic capacity is handled on finger rings from the second half of the 16th century.

Forget-me-nots are perennial, herb-like plants growing to about 25 cm in height (Figure 2). They are widespread throughout much of Europe, Asia and the Americas and prefer semishady, moist locations. Their flowering time ranges from May to September, depending on the location. Their flowers grow in clusters around a white-yellowish center and consist of five delicate, mostly blue petals. Along the stem rise several lanceolate, hairy leaves, to which the forget-me-not owes its botanical name *myosotis—mouse ear* (Plinius, 27, 80).

The forget-me-not has a long, not always unproblematic, tradition in the European culture of objects, souvenirs and gifts[1]. Particularly formative for the widespread perception of the forget-me-not has been its use on 18th century porcelain. During the Biedermeier Romantic period, the symbolic little plant was often placed on dedication cups, which were usually presented as birthday gifts and were intended to figuratively assure the recipient of the esteem shown to them with mottoes such as "*Friendship and trust*" or "*Your image* (rose) *My wish* (forget-me-not)" (Wiewelhove 2005, p. 39, 87, fig. 62, 93)[2]. The strong presence

of the "blue flower" (Blaue Blume) in the context of this time, which was oriented toward bourgeois ideals with a pronounced culture of friendship and love and the corresponding motifs of longing, may have led to the forget-me-not as a motif being associated today rather with negatively charged terms from the realm of pathos, nostalgia and kitsch[3].

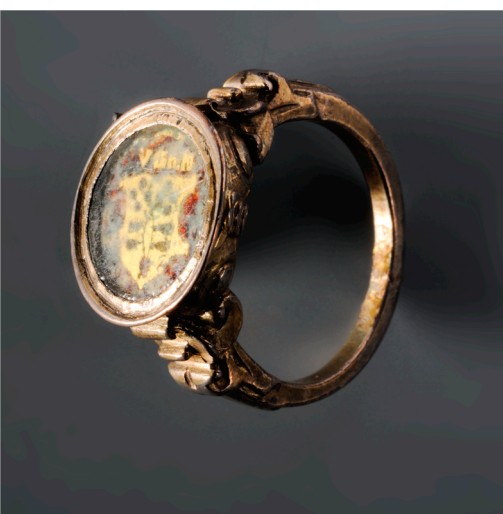

**Figure 1.** Finger ring with the device VGMN (Vergissmeinnicht), ca. 1575, gold, reverse glass painting, dm. bezel: 1.3 cm, Vitromusée Romont, Sammlung R. + F. Ryser, inv. Nr. RY 1012, ©Vitrocentre Romont, Yves Eigermann.

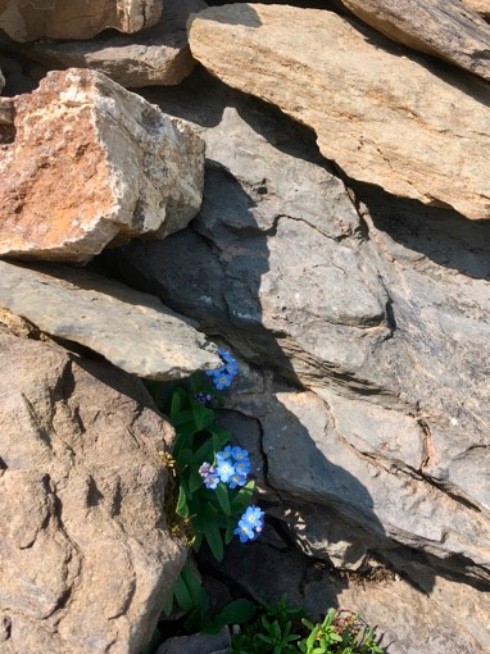

**Figure 2.** Blossoming forget-me-nots in August (Switzerland, Bundstock (2755 m)), ©Romina Ebenhöch, 2021.

It seems to be forgotten that the plant has been used as a carrier of meaning since the Middle Ages, and is represented in literature, painting and material culture. Before discussing in depth the use of the forget-me-not on finger rings, the first step is to create a cultural–historical framework in the sense of an approach by means of a painting, against which the use of the flower and its implied associations and qualities are called up. In its use as a pictorial motif in painting, it mediates between the real plant and its use on



actual objects, thus enabling a differentiated perception of the qualities of the forget-me-not within its individual manifestations. In addition, the following pictorial example already reveals a central aspect in considering the forget-me-not in Middle Ages and early modern art and cultural history, which lies in the constant presence of a message conveyed in image and text with sender and addressee.

ICH/PINT MIT/, VERGIS MEIN NIT

*I bind with for-get-me-nots*—Prominently placed in the picture, this statement is addressed to the viewer in a small-format painting by Hans Süss von Kulmbach, created around 1508 (Figure 3)[4]. The inscription is attached to a fluttering banner that winds around the central post of a fictitious window frame. In an Albertine manner, it oscillates between the actual picture frame and the painted window as a threshold into the picture (Alberti, de pictura, 19)[5]. However, contrary to the metaphor of the window, there is neither a view outward into a vast landscape nor into an interior space, but rather allows the viewer to become an observer of an intimate scene at this very threshold. Directly behind the windowsill, leaning against the frame to the left of the picture, a young woman with a beaded ribbon in her loose hair bound with a braid above her forehead, sits there weaving a wreath of forget-me-nots.

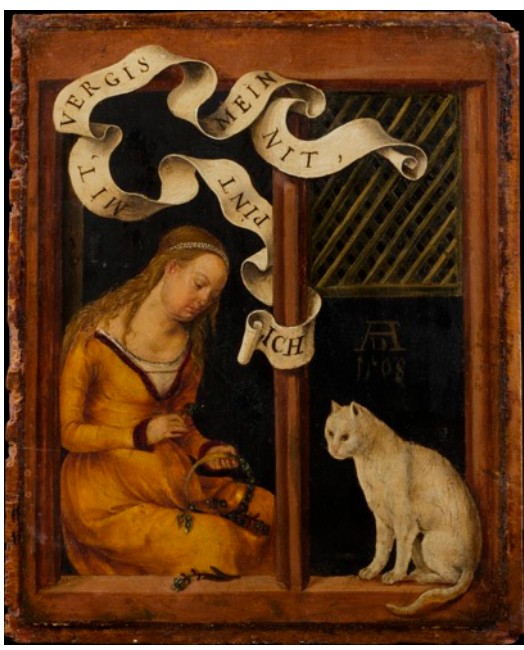

**Figure 3.** Hans Süss von Kulmbach, Girl Making a Garland, Nürnberg, 1508, oil on poplar, 17.8 × 14 cm, New York, Metropolitan Museum, 17.190.21, Gift of J. Pierpont Morgan, 1917, public domain.

The placement of the scene at the threshold of the painting in front of a blackened background, as well as the duality of the composition of the painting in two fields separated by the window bar, acquires a literal twist when one turns to the back of the painting, as seen in Figure 4. There, against a black background, is the portrait of a young man whose gaze is focused on a target outside the picture. In terms of pictorial structure, he occupies exactly the space filled on the opposite side by the window beam, with the banner winding around it.

While the depiction of the young man is largely unanimously regarded in research as portrait-like, the interpretations of the young woman fluctuate between portrait, genre painting and allegorical pictorial invention[6]. Within art history, double-sided paintings are rather rare, especially in the combination of portrait and allegory. According to Maryan Ainsworth, other contemporary examples include a portrait of a man attributed to Albrecht Dürer's circle with a depiction of Phyramus and Thisbe on the reverse, and Jacopo de'

Barbarai's portrait of a young man with a depiction of an unclothed pair of lovers on the reverse (Ainsworth et al. 2013, p. 166–171, 308–309).[7] A central aspect of this type of painting is that only one side, primarily the portrait side, is ever visible, while the other side, with its spectrum of meaning in the background, is only temporarily present or pictorially imaginable to viewers initiated into the mystery.[8]

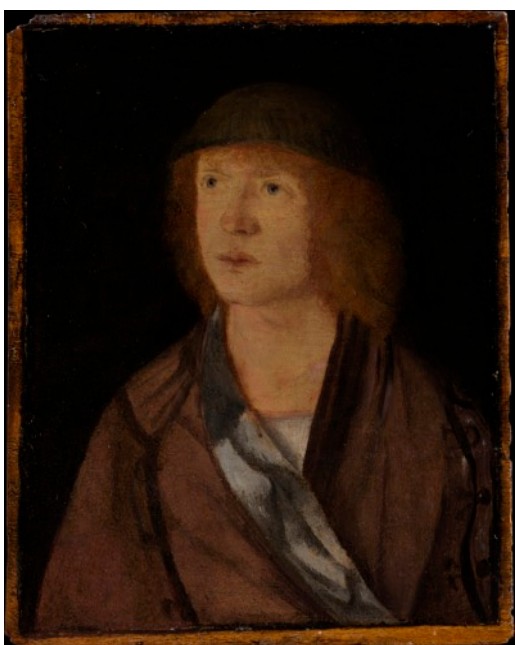

**Figure 4.** Hans Süss von Kulmbach, Portrait of a Young Man, Nürnberg, 1508, oil on poplar, 17.8 × 14 cm, New York, Metropolitan Museum, 17.190.21, Gift of J. Pierpont Morgan, 1917, public domain.

Unlike the two pictorial inventions of Dürer and de' Barbari, which show views in and out, Hans Süss von Kulmbach's depiction moves solely on the level of the window frame and thus merges the double-sided poplar panel as a material pictorial ground with the double-sided depiction. The inscription band, which moves alternately in front of and behind the window frame, is also integrated into this play. As a counterpart, so to speak, to the self-confident attitude of the pictorial form of the portrait, the self-assertive word "I" is placed prominently in the center on a linguistic level. In the rich network of relationships between the two sides of the picture, this I can, on the one hand, be related to the young woman, who allegorically refers to the creation of a connection in the sign of the wreath weaving (ICH/PINT MIT/VERGIS MEIN NIT). At the same time, however, it also allows us to read to the young man on the reverse and his longing gaze in the direction of the words VERGIS MEIN NIT. This ambiguity is further reinforced when one turns to the cat, which is interpreted as a symbol of the young man's loyalty and attentiveness, and whose gaze points to the figuratively transposed forget-me-not flower on the windowsill (Dittrich and Dittrich 2005, p. 259, 266). At the same time, in the cat's squinting at the plant, a hidden amusement with the word meaning of forget-me-not may also be intended. Unlike the German name Vergissmeinnicht (forget-me-not), the Greek name of the plant is based on the hairiness and shape of its leaves, which consequently give it the meaningful title *myosotis—mouse ear*[9]. This game with visual directions gains additional content for today's viewer if one adds the common name in Switzerland for the little plant as *"Chatzenäugli" (Cats Eye)* (Schweizerisches Idiotikon n.d., p. 875).

*The Forget-Me-Not as a Powerful Plant and Magical Motif in Medieval Literature*

While the Greek term *myosotis palustris* continues to be authoritative for the botanically correct and specifying designation of the plant from the violet genus, the name *Vergiss-meinnicht* has become established in the vernacular sphere. This name has been handed down in writing since the 14th century at the latest and can be found in all Germanic and Romance languages. Thus, the plant is called *forget-me-not* in English, *non-m'oublie(z)-pas* in French, and *non-ti-scordar-di-me* in Italian (Meyer 1951, p. 510).

In a 1951 essay, Herman Meyer followed the imperative of eleven German plant names, of which the forget-me-not has by far the widest distribution (Meyer 1951).[10] Moreover, it is the only plant whose imperative name structure is commented on in courtly literature of the Middle Ages.[11]

Thus, a Nuremberg collection manuscript from the second half of the 15th century, following a Teichner collection, contains a narrative in 166 verses whose title and content are dedicated solely to the little flower.[12] After a man's long search to find the name of the plant, the story ends with the words of a woman who is able to give him information and who asks him not to forget the noble little flower and to plant it in the garden of his heart: "*...Sy sprach vergismeinnit das edel pluemelein Pflantz mir in den garten des hertzen dein Vnd der zawn vmb den garten galt Soll sein nitliebbers vnd vergismeinnit an aller statt*" (she said, forget me not the noble flower, plant me in the garden of the heart and the fence around the garden shall be nitliebbers and for-get-me-nots everywhere).[13]

Already a half century earlier, Hans Vintler recommended in 1411 in "*die Pluemen der Tugend*" (Flowers of Virtue) in verse 8554 to talk to women "*of flowers forget-me-not and pretty minne*" ("*von pluemen vergismeinnicht und von hübscher minne*"). If the power of words was not enough, one could also make use of the magical power of the plant. Thus, in the 15th century, there is evidence for the idea that by carrying the plant, one should "*not be forgotten*" by one's "*dear*".[14]

The sources indicate that the forget-me-not was used mainly in the context of love spells. However, according to the dictionary of superstitions, it could also have apotropaic functions. Thus, one was supposed to be protected from all kinds of harm if one pricked out a wild forget-me-not with three spade pricks on June 24 (Bächthold-Stäubli and Hoffmann-Krayer 1987, p. 1568–1569). The idea of the effective power of the little plant had blossomed in all sorts of ways and caused the Wetterau pastor Conrad Rossbach in 1588 to take a humorous–critical position on it in his *Paradeißgärtlein*, writing:

> "*diss kräutlein Art und eigenschaft. Nicht viel man find soll geben kraft—den bulern und sie machen werth, den weibern, also gar verkehrt. Sindt abergläubisch Leut fürwar Und hilfft doch oftmals nit ein Haar.*" ("The herb's nature and properties are not much to be found—they give strength to the wooers and make them worthy of the women, so quite wrong. These are superstitious people indeed and yet often it doesn't help anything.") Conrad Rossbach, Paradeisgärtlein, 1588; quoted from (Bächthold-Stäubli and Hoffmann-Krayer 1987, p. 1569).

Conrad Rossbach's critical comment on the use of the forget-me-not is of particular importance in the context of what follows, as it points to the deeply rooted understanding and widespread practice of using the plant as an effective and emblematic object around 1588. At the same time, the numerous different legends seem to have formed the breeding ground by means of which the particular prevalence of the forget-me-not as a pictorial motif in the second half of the 16th century and at the beginning of the 17th century becomes understandable.

## 2. Sealing Love, Friendship and Closeness to God: The Forget-Me-Not in Connection with Rings in the 16th Century

In the course of the 16th century, not only literary sources can be cited that report on the word power of the little flower, but also figurative realizations that play with the appellative language ability of the forget-me-not. The forget-me-not, as in Hans Süss von

Kulmbach's double-sided painting, can be integrated into allegorical references that revolve in particular around the element of connection in the weaving of a wreath of forget-me-nots. It is already significant in this example that the little plant is tied to an object that is already strongly charged with the wreath itself.[15] At the same time, the painting's construction as a reversible image draws attention to another aspect, which lies in the presence of sender and addressee on three levels (language, image, object). The forget-me-not, as a real plant as well as a linguistic or pictorial motif, always points beyond itself and carries with it, with a pronounced power of reference, an imagined counterpart. This counterpart, as seen in what follows, can be understood and occupied in different ways.

At the latest in the 16th century, the forget-me-not motif can be found on rings, on which it enters into a strong connection between image and image carrier with its imperative linguistic competence. As objects, rings have an almost inherent power of reference. They are among the most widespread objects of gifts and a gift exchange that reaches back to antiquity and is still practiced today.[16] Similar to the forget-me-not as a motif, they are highly charged as objects. In numerous medieval romances, they appear as magical protagonists and essential components for the happy or dramatic course of a story (Graf 2019, p. 163–176). They were given and sent as New Year's gifts and presented as pledges, expressions of loyalty, proof of fidelity or gifts of love.[17] They could not only seal marital unions, but as signet rings they could also assume legally binding functions (Scarisbrick 2007, p. 28–44). In order to do justice to these different functions or to convey them, rings carry a rich spectrum of inscriptions and pictorial motifs.

In addition to pictorial motifs such as the *dextarium iunctio* or *Fede motif* as a sign of a marital union, the heart as a proof of love, or the skull as a *memento-mori* motif, it is especially engraved inscriptions that function as bearers of the defined message. Powerful names and words such as "*IHS*" or "*Tetragrammaton*" can offer protection, while sayings such as "*Thinke wel of me*" or "*I like my choyce*" convey proof of favor.[18] In the context of these numerous different motifs and inscription forms, the forget-me-not occupies a special position, insofar as it combines word and image, making the inscription itself a pictorial sign.

So far, at least nine rings can be traced on which the forget-me-not appears (Figure 1). In most cases, the rings are finger rings with a diameter of about 2.5 cm on which a circular ring head with an average diameter of 1.3 to 2.2 cm is attached. The ring head holds a slightly convex rock crystal or glass inlay on which the motif is depicted in gold foil on a translucent red background using a special technique of reverse glass painting known as amelioration.

Common to all of them is the depiction of a coat of arms with three forget-me-not flowers together with the abbreviation of the plant name as VGMN, VMN or also FGMN. The different use of the spelling FGMN speaks for the fact that the rings were widespread not only in southern Germany, but also in England under the English name "forget-me-not". In most cases, the heraldic shield of the rings with a round bezel flanks a date, which dates the objects between 1562 and 1583. In some cases, the ring heads represent the combination of two heraldic shields in the style of an alliance coat of arms, which add the figurative name of Jesus *IHS* to the linguistic floral representation.[19]

In most of the objects, the glass surface is additionally incised to reinforce the motif of the heraldic shield with the flowers in the glass cut (Figure 5). Thus, the glass cut takes up the design of signet rings and allows the ring to be pressed into wax similar to a seal, and thus to be understood in the sense of a pseudo signet ring.[20]

Signet rings in reverse glass painting have been known since the second quarter of the 15th century and had their greatest spread from the first half of the 16th century, especially in southern Germany, but also in England and the Netherlands (Figure 6). In addition to the heraldic shield, the rings are usually authenticated by a year and the owner's initials.[21] In imitation of legally valid signet rings, the forget-me-not rings make their formal and pictorial language their own by means of a glass cut, heraldic shield and initials. In doing so, they metaphorically transform the idea of individual sealing to a universally valid sealing

of relationships and place this with the coat of arms and the initials of the forget-me-not under the service of a call to commemoration.

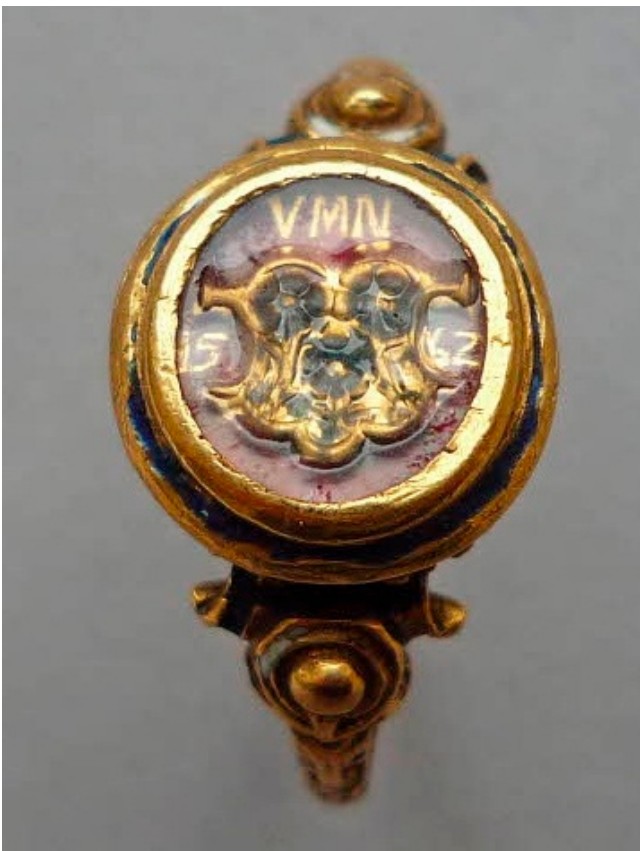

**Figure 5.** Pseudo-Signet-ring with the device "*VMN*" and the year "1562", Gold, rock crystal, reverse glass painting, dm. bezel: 1.1 cm; dm. ring: 2.1 cm, London, British Museum, AF.643, ©The Trustees of the British Museum, (CC BY-NC-SA 4.0).

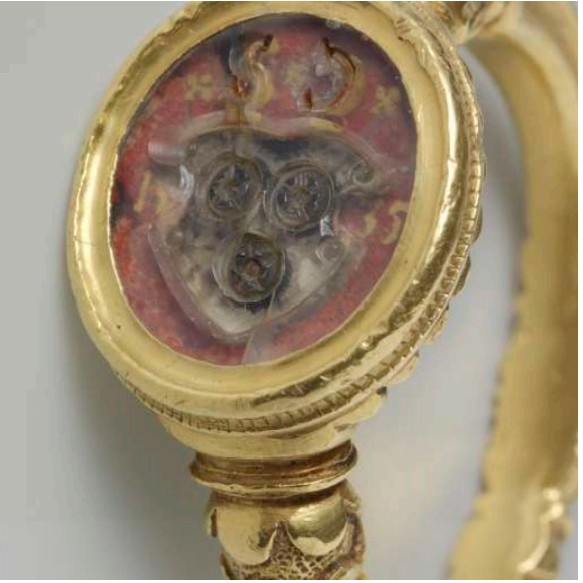

**Figure 6.** Signet ring with the Van Wijnbergen family arms "*CS/1535*", Northern Netherlands 1535, gold, rock crystal, dm. 3 cm, Amsterdam, Rijksmuseum, BK-NM-9567, public Domain.

But whose memory is to be planted in the heart of the ring bearer? Especially the sources mentioned at the beginning about the use of the flower in courtly love poetry (minnesong) or in practices of love magic support the relational, respectively manifesting dimension of the forget-me-not in the context of a connection between lovers or spouses. The small-format, double-sided painting from the Metropolitan Museum suggests something similar, on the front and back of which the portrait of a man and the depiction of a young woman tying a wreath of forget-me-nots, as a symbol of a connection, are relationally linked. The rings themselves may allude to this theme in the form of the alliance coats of arms, which also place the union under the protection of God through the monogram of Christ *IHS* on the left.[22]

The tradition of using rings as a sign of loyalty and eternal fidelity in the context of a marital union dates back to antiquity. Depending on religion, region and era, rings were incorporated into the rite of marriage or tied to specific conventions of wearing.[23] Solid evidence for a use of the forget-me-not on rings as actual wedding bands can only be traced to the 17th century. A ring from the Victoria and Albert Museum attests to the act of marriage through several components: for example, it lists two pairs of initials, gives the exact date, 12 June 1634, and combines the little flowers with the depiction of the *Fede*-motif, the so called "*mani in fede*" or "*dextrarum iunctio*".[24]

A further indication for the use as a gift of love is the rings' proximity to a group of objects that is also situated on the finger, but belongs to a completely different context of use. Thimbles in their close use on the finger of the hand connect the symbol of the forget-me-not flower to the symbolically charged "handmade" creation of a wreath. Interestingly, identical round reverse glass inlays are found on the tops of Nuremberg thimbles of the same period [25]. Accordingly, eleven thimbles with the coat of arms and initials of the plant name as a motto VMN can be added to the nine surviving rings. The elaborate thimbles were considered masterpieces of Nuremberg craftsmanship and are interpreted as valuable wedding gifts from the husband to the future wife.[26] Their shafts often bore additional mottoes, including "*what god gives lasts forever,*" "*alone mine or let be,*" or "*heartfelt love never parts*".[27]

Although the use of the forget-me-nots unfolds a strong plausibility in the named contexts, it is not exclusive to them. Significantly, the only surviving source reports the gift of such a ring not as a gift exchange between husband and wife. In the chronicle of Brno by the councilor and apothecary Georg Ludwig, transmitted by P. Chlumecky in 1859, the entry for the year 1582 reads: "*On July 8, Christoff Chirmesserus, pastor of St. Jacob's, gives me a ring 'vergis mein nit'*".[28]. In 1582, a ring with the motif of the forget-me-not was enough of an appropriate gift between two men to be recorded in a chronicle. What kind of relationship was attached to it is just as difficult to comprehend today as the kind of commemoration that was associated with it. However, the fact that the ring was given by a cleric should give the occasion to look at the rings from a further and final perspective.

To this end, we should once again refer to the already mentioned combination of the forget-me-not with the Christ monogram *IHS* in the style of an alliance coat of arms, which can be proven so far on two rings and three thimbles.[29] It is remarkable that the otherwise given three flowers are now limited to two flowers with the addition of the *IHS*. Apotropaic ideas are connected with the sign *IHS*, which place the wearer of the ring under the Christian–magical protection of the sign.[30] Thereby, the alliance coat of arms connects the name of Christ by the representation of the forget-me-nots with a request for remembrance. The fear of forgetting the name of God is especially connected with the mystic Seuse (14th century), who went so far as to tattoo it on his chest with a writing stylus, as it were in maximum body proximity (Bihlmeyer 1961, p. 15-16). Although clearly less invasive, a similarly close body proximity is offered by pieces of jewelry with the monogram, frequently encountered between the 15th and 17th centuries.[31]

In the Middle Ages and the early modern period, memory processes in the sense of *recordatio* or *taking to heart* are strongly connected with the heart as the seat of the soul.

Ideally, good images are to be inscribed in it or imprinted in wax, like a seal.[32] Especially, the metaphor of the seal gains an additional dimension, if one connects the act of impressing or imprinting in wax with the appellative commemorative request of the forget-me-not. At the same time, however, it is inherent in the object itself, when the glass cut refers to its function as pseudo-sealing rings and represents a (pictorial) material to be imprinted in wax. In this sense, the rings serve established ideas and topoi of memory processes, and even extend them.

In addition to the metaphor of inscription and imprinting, the image of planting in the garden of the heart, as it appeared in the verse narrative cited at the beginning, served as a reference[33]. The planting of the forget-me-not in the garden of the heart mentioned there, finds a direct pictorial conversion in the rings and is additionally strengthened by the blood-red background. This is particularly evident on a rosary pendant with the written form of the "forget-me-not", on which the implementation of the motif with the heart and the sprouting plants could be more detailed due to the available space than on the small-format ring heads (Figure 7).

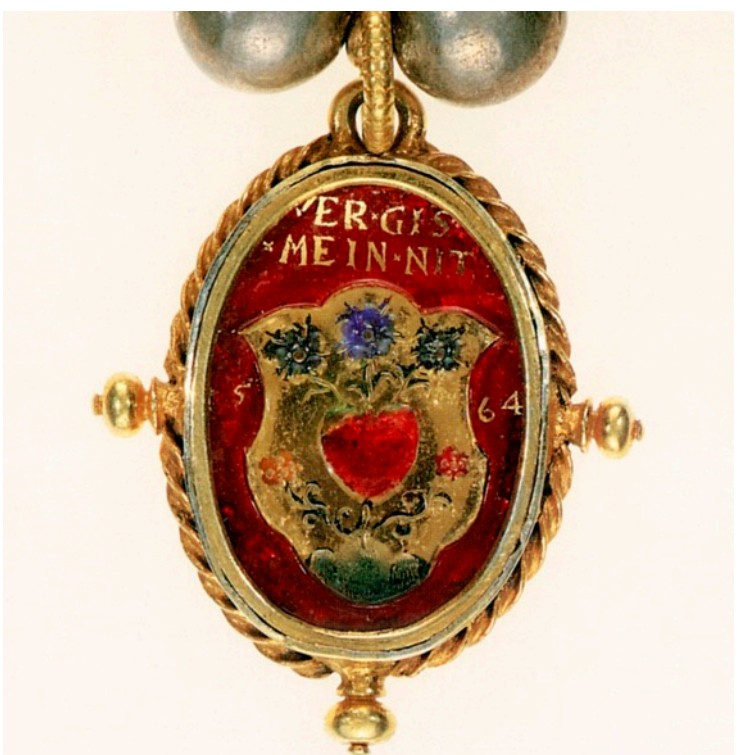

**Figure 7.** Rosary pendant, "Vergis mein nit 1564", Nürnberg, 1564, Bayerisches National Museum, München, Inv. Nr. 76/90 (formerly), depicted in: (Ryser and Salmen 1995), Kat. Murnau 1997, 110, Cat. N0. F15.

The potential of jewelry objects to activate memory "in the sense of reminiscent envisioning and taking to heart" („im Sinne von erinnernder Vergegenwärtigung und Beherzigung") had occupied Silke Tammen in a 2015 essay on a "Memanto" ring, the so-called Thame ring (Tammen 2015, p. 299–324). Similar to the forget-me-not rings, the 14th century ring addresses its addressee with the inscription "*Memanto Mei Domine*".[34] The call to "*Memanto*" on the ring plate is addressed to the wearer, as well as in the surrounding continuation "*mei domine*" to God. The forget-me-not rings also aim at this double meaning of mutual remembrance. The communication of the Thame ring, however, is limited to a communication between the wearer of the ring and God. In the case of the forget-me-not rings, a third part is added to this connection. The creation of a thimble from the Wellby



Collection brings this, if you like, "triangular relationship" to the concise formula "*God and yours shall be unforgotten 1573*". "[35]

In this sense, the objects are able to call up several connections at the same time and allow God to appear as an imagined part of a commemoration, which can be directed to spouses as well as to valued, close or missing persons.

In the rings, multilayered functions intermingle, which are due to the expression or desire for connection, the different facets of commemoration, the desire for divine assistance, but also the belief in the power of speaking protective formulas in words and images.

**Funding:** This research received no external funding.

**Conflicts of Interest:** The author declares no conflict of interest.

## Notes

[1] The German War Graves Commission (Volksbund Deutsche Gräberfürsorge) has chosen the forget-me-not as the symbol of its campaign "100 Years of World War I. Against Forgetting", https://www.100-jahre-erster-weltkrieg.eu/gegen-das-vergessen.html (accessed on 20 February 2022). According to (Geppert 1996, 110) the forget-me-not is said to have been a sign of the Freemasons during the time of National Socialism.

[2] "*Freundschaft und Vertrauen*"; "*Dein Bild* (Rose) *Mein Wunsch* (Vergissmeinnicht)".

[3] For more information on the symbol of the Blue Flower in Romanticism see (Hecker 1931).

[4] The initials AD for Albrecht Dürer are shown to have been added secondarily, based on research at the Metropolitan Museum (Ainsworth et al. 2013, p. 166–171, 308–309).

[5] On the threshold see (Bawden 2014).

[6] On the research discussion on European Portraits see (Ainsworth et al. 2013, p. 166–171, 308–309). An analogy could possibly be drawn on an allegorical level to the love spell in Leipzig, Lower Rhine Master, Der Liebeszauber, painting, Germany, c. 1470/1480, oil on wood, 24 × 18 cm, Leipzig, Museum der bildenden Künste, Inv. Nr. 509.

[7] Jacopo de' Barbari, Double Painting with Portrait of a Man and an unclothed lovers pair, Italy, c. 1455–1516, oil on wood, 60.7 × 45.6 cm, inv. no. 1664, Berlin, Gemäldegalerie, Berlin; Albrecht Dürer (succession), double painting with portrait of a man and Pyramus and Thisbe, Franconia, c. 1515, oil on wood, 34 × 27.4 cm, Colmar, Museum Unterlinden, Colmar, inv. no. 87.1.1.

[8] In contrast to the genre of painting, a double-sided and meaning-linked use of an image or text carrier is extremely common in jewelry culture.

[9] Around the small flower with the delicate blue blossoms entwine numerous legends, together with attributions of magical powers and qualities. In Baden and Bohemian Forest, the plant is also said to have been called the blue sky key. In traditional legends, it is said to have forgotten its name after the creation of the world, or to be able to point out the place where treasures are found (Bächthold-Stäubli and Hoffmann-Krayer 1987, p. 1568–1569).

[10] In addition to Vergissmeinnicht, Meyer addresses ten other plant names, including Krückche-rier-mich-net-an (impatiens noli tangere), Macht-heil (senecio sarracenicus), Schmecke-nicht (ipomea jalappa), or Nimm-mir-nichts (alchemilla alpina) (Meyer 1951, p. 509–516).

[11] According to Meyer however, this circumstance could also result from the findings (Meyer 1951, p. 509).

[12] Unknown, verse narrative *Von dem Bluemlein Vergismeinnit*, second half of the 15th century, surviving in a manuscript of the British Library, London, BL Add. 24946, 53r-55r. (Klingner and Lieb 2013, p. 576–577).

[13] The textual allegory recalling the garden of the heart is thereby also associated with the image of the enclosed garden and the inner garden of virtues planted by God (Falkenburg 1994).

[14] "*ein blumelin heiszet vergisse myn nit, dem des empholen wird, der magk woele frohlichs muts sin, der isz von ime selber dregt, der wiele sins liebs nit vergessen zu keiner zit.*" Delivered according to (Grimm, Altdeutsche Wälder, 1, 151).

[15] According to Brahm flower wreaths were considered, among other things, as "headdresses and badges of honor of the virgin bride" (Brahm 1942, p.1125–1130). The closed ring shape was a symbol of virginity, which is why they were discarded forever on the wedding night. The custom ties in with the Germanic custom of virgins' hair falling open and held in place by a band of flowers and pearls. See also (Vavra 1991, p. 1475).

[16] On the history of rings see (Hindman 2015; Scarisbrick 1993; Scarisbrick 2007).

[17] On the exchange of gifts and jewellery see (Hirschbiegel 2003; Scarisbrick 2007, p. 55–83).

[18] On the function and meaning of inscriptions see (Skemer 2006, p. 75–124; Hindman 2015, Cat, No. 15, 40; Scarisbrick 2022).

[19] Pichtures of Finger ring, 'FGMN', Germany, late 16th century, gold, rock crystal, reverse glass painting, 2.2 × 2 × 0.9 cm, London, V&A, 815–1871. (https://collections.vam.ac.uk/item/O120994/signet-ring-unknown/?carousel-image=2015HN6235). (accessed on 23 August 2023). See also (Battke 1954, p. 61–62, Nr. 77; Oman 1930, Nr. 602).

[20]   Pseudo signet rings imitate signet rings with their specific features without having their actual function. On the use of Pseudo signet rings see (Stürzebecher 2014, p. 66–67).

[21]   Signet rings in reverse glass technique were used by high dignitaries and the clergy as well as by persons of the upper middle class. See also, in addition to Figure 6, the signet ring of Duke Maximilian of Bavaria "MHIB", c. 1600, gold, rock crystal, amelioration, dm. 2 cm, Vienna, KHM, Kunstkammer, Antikensammlung, XII 313.

[22]   See Note 19.

[23]   On the use of rings in marriage contexts see (Scarisbrick 1988–2021; Stürzebecher 2020).

[24]   Pictures of Finger ring 'AW + GH/ANNO/1634/12 IVNI', Germany, 1634, gold, glass or rock crystal, reverse glass painting, 2.4 × 2.5 × 2.6 cm, London, V&A, M.229-1975. (https://collections.vam.ac.uk/item/O119801/signet-ring-unknown/). (accessed on 23 August 2023).

[25]   Pictures of Thimble, *VMN*, Nürnberg, second half of the 16th century, silver-gilt, glass, reverse glass painting, 1.8 × 1.5 cm, Oxford Ashmolean Museum, Inv. Nr. WA2013.1.91 1573, ©Ashmolean Museum, University of Oxford. (https://collections.ashmolean.org/collection/search/per_page/25/offset/0/sort_by/relevance/object/68038). (accessed on 23 August 2023).

[26]   Nuremberg thimbles of the 16th century were the subject of the collector couple Isbister (Isbister and Isbister 2014).

[27]   Object details: Private property: VMN 1580 "was got bescherdt imer werdt" 1579; Formerly Figdor collection: "Herzlich Liebe scheid sich nie 1582"; "Allein mein oder las gar sein" (Zander-Seidel 2015, p. 218).

[28]   "*Den 8. July schenkt mir Herr Christoff Chirmesserus Pfarrherr zu S. Jakob einen Ring vergis mein nit*".

[29]   See Note 19.

[30]   On apotropaic aspects see (Skemer 2006, p. 75–124; Jones and Olsan 2000, p. 256–268)

[31]   Pendant with *IHS* and Arma Christi, 1580–1600, gold, enamel, diamonds, 6 × 3.6 cm, London, V&A, inv. no. M248-1923. They made it possible to wear the name of Christ close to the body, to assure oneself of the constant visualization and remembrance of the name, and to keep it pictorially present in the wearer's memory.

[32]   On metaphors of memory see (Carruthers 2009, p. 16–32, 33, 49, 62, 72; Tammen 2008, p. 106–107).

[33]   See Note 12.

[34]   Thame Ring, England, 14. Jh. '*Meman(t)o'/, Mei Domine*', Gold, Amethyst, 2.5/1.6 cm, Oxford, Ashmolean Museum, Inv. Nr. AN 1940.228. For images see (https://britisharchaeology.ashmus.ox.ac.uk/highlights/thame-hoard.html). (accessed on 23 August 2023).

[35]   "*Gott und dein soll unvergessen sein 1573*". See Note 25.

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
