# Peer review of "“Vergis Mein Nit”—Connectedness and Commemoration through Rings in the 16th Century"

_arts_

Round 1
Reviewer 1 Report
This is a most interesting article on nine rings with a particular symbol (the forget me not). The background to the use of the symbol is fully explored, and this preliminary part might well be shortened slightly.
I was disappointed that the nine rings are not listed and not described in greater detail. Line 241 has the footnote number 27 but no text listing the rings is given for this footnote. While a time frame is given for their production 1562 to 1583, there does not appear to be any discussion of where they were produced. Are they all German?.
Fig 7 also is dated 1634, well outside the initial time frame. This needs more comment.
In line 145 'chatzen augli' needs some explanation for those unfamiliar with Swiss German.
Line 175/6 the suggestion that by carrying the plant or one might not be forgotten by one's dear might be linked to its use in rings.
Also there is no suggestion of which country Romont is in. There appear to be Romonts in France, Switzerland and the USA.
There needs to be more explanation of what a pseudo-signet ring is. The BM online collections website for this ring (AF.643) simply describes it as a signet ring.
The English is in general quite clear. There is some inconsistency in the use of capitals, and this should be corrected according to the editorial rules of the publishers. There also seems to be confusion between foot or end/notes and the Harvard system.
In line 276 I do not understand the phrase 'minned poetry'.
In line 42 the commission should be translated as the German war graves commission.
In line 89 the phrase 'half-opened hair' might be better expressed as 'loose hair bound with a braid above her forehead'. And, while on hair, line 212 falling 'loose' rather than 'open'.
Line 226 'romances' rather than 'novels'.
Line 328 'translated' might be better than 'transmitted'. But we are not told what the original language is. Was it Czech or German?
Author Response
Thank you for your review. Please check the attachment.

Reviewer 2 Report
For the most part this paper is sound in terms of argumentation and organization, engagement with the previous literature, and evidence/sourcing, and I enjoyed learning about this aspect of northern European memorial culture. However, there are many small proof-reading and consistency issues ("forget-me-not" vs. "forget me not," for example) and oversights in completing translation of the footnotes (Abb. instead of Fig., Kat. Nr. instead of Cat. No.).
Some specific notes addressed to the author:
Line 107 -- clarify that you are talking about just northern portraits, or if not I would suggest bringing in the Ginevra de' Benci portrait by da Vinci.
Line 169ff -- translate the German passage
Line 307ff -- could a connection to be drawn between the forget-me-not appearing on thimbles [embroidery/sewing/weaving] and the fingers creating a wreath of forget-me-nots? the correlation between the motions of the thimble-wearer and those of the wreath weaver (as in Kulmbach's painting) might explain their presence there
Throughout -- any time you use the verb "seems," see if you can instead find a way of saying it with more conviction. English-language scholarship interprets the term "seems" (scheint or wirkt) as the author lacking confidence in the conclusions they are drawing, which I am sure is not the intention :)
It is clear that this paper has been translated from German into English, and not always very successfully. Many sentences are fine, while others are quite wordy and convoluted. Some suffer from overuse of the passive voice, which is common in German but far less so in scholarly English in the humanities. Many sentences betray the original German grammatical organization of the sentence remaining in the English translation--"A similarly close, although clearly less invasive body proximity offer pieces of jewelry..." (347-348). Sometimes there are issues with mis-translations or usage issues--for example, "object culture" in line 62 in English is usually called "material culture," "minned poetry" in 282 should be minnesong (we use the German term), and "half-opened hair" (halb-offen) in line 89 would be better as "half-up." I'd encourage the author to liaise with a native English speaker to assist with clarity and flow of individual sentences--Lance Anderson in Berlin specializes in art-historical translations and may be a resource the author could pursue.
Reviewer 3 Report
This is a well-researched essay with a sophisticated and nuanced argument. It deserves publication.
In the abstract and elsewhere, change the spelling of coat of arm into “coat of arms.”
In the abstract: in “Instead of an actual coat of arm though these finger-rings carry the Device V(G)MN or FGMN (For-get-me-not) accompanied by a depiction of little blue forget-me-not-flowers as coat of arms” add a comma after the word “though.”
I am not sure what is the format of the journal for references to the size of objects. In the captions, the author notifies the size of the ring 3,2 cm. I wondered if it should be described in the format of 3.2.
Note 13 should be corrected. Meyer, Herman C. "The Imperative in German Popular Plant Names." The Journal of English and Germanic Philology 50.4 (1951): 509-516. The book’s title is not fully written while sometimes the author inscribes “imperative” instead of “imperative.”
I advise the author to refer somewhere after line 203 to the visual and textual allegory of the “enclosed garden” and the “inner garden of virtues,” associated with God planting the virtues in the soul of the believer. In the commentary of Richard of St. Victor on the enclosed garden in the Song of Songs, for example, the garden appears as an inner locus, a place where the soul can spiritually taste the plants of virtue. s/he can consult Reindert Leonard Falkenburg, The Fruit of Devotion: Mysticism and the Imagery of Love in Flemish Paintings of the Virgin and Child, 1450–1550, trans. Sam Herman (Amsterdam and Philadelphia: John Benjamins, 1994), as a starting point.
Line 289: The author should explain in which part of the ring we can find the monogram of Christ HIS. The author is also advised to draw a circle around the specific place in the illustration.
The paragraphs concluding the essay are very short. After such a sophisticated and nuanced discussion, I would appreciate more expended conclusions. One point that emerges from the article and deserves consideration is the fact that the commercial circulation of rings in a capitalist society assisted in embracing diverse meanings of the aforementioned rings.
Round 2
Reviewer 2 Report
the following are all formatting, capitalization, spelling, and consistency issues, with reference to the line numbers
5 monogram
6 coat of arms
8 device
10 remove hyphen in finger-rings for consistency
12 remove hyphen in for-get
12 power
13 German
14 jewelry (“jewellery” is the UK spelling, but the spellings in the article are all US)
24 replace “typically in” with “typical of”
25 device
38-39 Note 1 is in the body of the text – needs to be moved to foot/end
41 Remove hyphen in For-get
44-49 Note 2 is in the body of the text – needs to be moved to foot/end
54-55 Note 3 is in the body of the text – needs to be moved to foot/end
60 Note 4 is in the body of the text – needs to be moved to foot/end
80-82 Note 5 is in the body of the text – needs to be moved to foot/end
85-86 Note 6 is in the body of the text – needs to be moved to foot/end
105-109 Note 7 is in the body of the text – needs to be moved to foot/end
110 Insert “Maryan” before Ainsworth
113 de’ Barbari
114-118 Note 8 is in the body of the text – needs to be moved to foot/end
121-123 Note 9 is in the body of the text – needs to be moved to foot/end
124 de’ Barbari
138 Note 10 is in the body of the text – needs to be moved to foot/end
142-147 Note 11 is in the body of the text – needs to be moved to foot/end
149-150 Note 12 is in the body of the text – needs to be moved to foot/end
156 Romance
157-158 Note 13 is in the body of the text – needs to be moved to foot/end
160-164 Note 14 is in the body of the text – needs to be moved to foot/end
165-166 Note 15 is in the body of the text – needs to be moved to foot/end
169-171 Note 16 is in the body of the text – needs to be moved to foot/end
179 remove highlighting on garden
180 remove highlighting on enclosed
183-184 Note 17 is in the body of the text – needs to be moved to foot/end
186-189 Note 18 is in the body of the text – needs to be moved to foot/end
193-194 Note 19 is in the body of the text – needs to be moved to foot/end
199-203 Note 20 is in the body of the text – needs to be moved to foot/end
219-224 Note 21 is in the body of the text – needs to be moved to foot/end
235-236 Note 22 is in the body of the text – needs to be moved to foot/end
238 Note 23 is in the body of the text – needs to be moved to foot/end
240 Note 24 is in the body of the text – needs to be moved to foot/end
242 Note 25 is in the body of the text – needs to be moved to foot/end
248-249 Note 26 is in the body of the text – needs to be moved to foot/end
252-253 Note 27 is in the body of the text – needs to be moved to foot/end
265-266 Note 28 is in the body of the text – needs to be moved to foot/end
272 Note 29 is in the body of the text – needs to be moved to foot/end
281-284 Note 30 is in the body of the text – needs to be moved to foot/end
302 I in IHS should be italicized also
302 The superscript “2” confusing, given the formatting issues of the other footnotes. Should be integrated into all foot/endnotes. Also, usually the superscript number goes immediately after the period at the end of the sentence.
305-306 Note 31 is in the body of the text – needs to be moved to foot/end
311 See note about superscript numbers above for line 302.
312 rings’
315 flower
317 See note about superscript numbers above for line 302
320-321 Note 32 is in the body of the text – needs to be moved to foot/end
323-325 Note 33 is in the body of the text – needs to be moved to foot/end
326 forget-me-nots
331-332 Note 34 is in the body of the text – needs to be moved to foot/end
333 suggest rewording to “…forget-me-not was enough of an appropriate gift…”
339 above in line 302 IHS is italicized, whereas here it is not; should be consistent
343-344 Note 35 is in the body of the text – needs to be moved to foot/end
348 Note 36 is in the body of the text – needs to be moved to foot/end
350-352 Note 37 is in the body of the text – needs to be moved to foot/end
357-358 Note 38 is in the body of the text – needs to be moved to foot/end
367 Note 39 is in the body of the text – needs to be moved to foot/end
370 comma should be before the quotation mark: “forget-me-not,”
372 elsewhere in the essay figure numbers are denoted by (Fig. #) rather than (Figure #); should be consistent
376-377 Why is this quotation in italics?
378-379 Note 40 is in the body of the text – needs to be moved to foot/end
381-382 Note 41 is in the body of the text – needs to be moved to foot/end
386 remove hyphen between not and Rings, also lower case r: forget-me-not rings
389-391 Note 42 is in the body of the text – needs to be moved to foot/end
461 In English we call these “Primary Sources”
The language is still clunky and awkward in places, as it still does not read like a native-language composition but as a translation. However, some improvements have already been made that help somewhat with the clarity and flow.
Author Response
Thank you!
